# Host plants selection of *Centranthera grandiflora* Benth. and nontargeted metabolomics analysis of its parasitic and non-parasitic samples

**Song Jin, Yuchuan Li, Jun Ni, Haili Xie, Falin Lei, He Liu**⊙*

Yunnan Characteristic Resource Plants Intelligent Agriculture Engineering Center, College of Agriculture and Life Science, Kunming University, Kunming, China

* britney198601@163.com

**Data Availability Statement:** All relevant data are within the manuscript and its Supporting Information files.

## Abstract

According to the previous investigation and research of our group, it was found that *Centranthera grandiflora* Benth. (*C. grandiflora* for short) might be a root hemiparasitic plant. The experiments of mixed sowing of *C. grandiflora* and 9 companion plants that might be hosts were conducted, and the growth, biological yield and other indexes were observed. The results showed that *Cyperus iria* L. was the best host for *C. grandiflora*, and when they were mixed sowed, *C. grandiflora* had a vigorous growth above ground and the haustoria connected obviously below ground, while *C. grandiflora* could achieve blossoming and fruiting in the same year. Next, nontargeted metabonomics analysis methods were utilized to clarify the differences in metabolites between parasitized and non-parasitized *C. grandiflora*. A total of 82 metabolites with significant differences were screened. The main upregulated differential metabolites of non-parasitized plants were for plant growth, while that of parasitized plants were functional compounds such as EPA. Out of 82 differential metabolites, 32 were annotated into 37 KEGG pathways. Analysis of the 37 pathways in combination with the differential metabolites showed that in addition to being involved in the synthesis pathway of iridoid terpenes, the up-regulated metabolites of parasitized plants were involved in the synthesis pathways of several functional components, while that of non-parasitic plants were involved in the subsequent catabolism of monoterpenoid compounds, as well as the metabolic pathways of nutrients synthesis and catabolism, energy generation, and phytohormone production required for plant growth.

## Introduction

*Centranthera grandiflora* Benth. is the most important primary plant of red rooted wild broad bean, belonging to the family *Scrophulariaceae*. It is a rare medicinal herb among the Miao, Yi, and Lahu ethnic groups in Yunnan Province [1, 2], and often used in the treatment of amenorrhea and dysmenorrhea, pediatric hyperthermia, infertility, traumatic injury and other

**Funding:** This work was supported by the Regional Grant of National Natural Science Foundation of China (Grant number 32360098); Youth Grant of National Natural Science Foundation of China (Grant number 32001684); Key Project of Yunnan Local University Joint Fund (Grant number 202201AN070036) and General Program of Yunnan Fundamental Research Projects (Grant number 202201AT070020).

**Competing interests:** The authors have declared that no competing interests exist.

diseases. It shows great prospects in the development of new drugs for cardiovascular diseases and leukemia treatments as well as healthy foods with antihypertensive, hypoglycemic or hepatoprotective effects [3]. Therefore, it is of far-reaching significance to solve the problem of artificial cultivation of *C. grandiflora* and explore its medicinal value.

The preliminary study of our group found that in the absence of a suitable host companion plant, *C. grandiflora* could not grow and survive alone for a long period of time, let alone complete a life cycle in the case of seedlings, seedling in culture, or 2-year-old rooted seedling with soil ball [4]. The leaves of *C. grandiflora* could undergo certain photosynthesis, and the protruding nodules on the roots were intertwined and connected with the roots of other plants around them. Preliminary microscopic examination of the nodule showed that its structure was similar to the haustorium of *Thesium chinense* Turcz [5] or *Pedicularis kansuensis* [6], that is, there were typical parasitic-plant organs, haustoria, in the roots of *C. grandiflora*. Thus, it could be inferred from above that *C. grandiflora* is a kind of root hemiparasitic plant. Based on the collected evidences of root hemiparasitism and field investigation results of companion plants species, 9 possible host plants were mixed sowed respectively with *C. grandiflora*, and the optimal host plant was selected by measuring various growth indicators of *C. grandiflora*.

In addition, the parasitic mechanism of *C. grandiflora* is currently unclear. Previous studies on endophytes have shown that there were material exchanges between *C. grandiflora* and its host [7]. It could be speculated that the host might provide *C. grandiflora* with special substances which are not commonly present in *C. grandiflora* and could not be obtained directly from soils. The substances could be the metabolites of the host plant, and might directly affect the efficacy components of *C. grandiflora*, such as iridoid glycosides. Therefore, in order to gain a more detailed understanding of the differences in metabolites between parasitized and non-parasitized *C. grandiflora*, LC-MS and nontargeted metabolomics techniques were utilized to analyze the impact of the optimal host on the metabolism of *C. grandiflora*. Multiple statistical analysis methods such as principal component analysis (PCA), orthogonal partial least squares discriminant analysis (OPLS-DA), pathway enrichment, and cluster analysis were used to explore the influence of the optimal host on the metabolites and metabolic pathways of *C. grandiflora*, and to elucidate the root hemiparasitic mechanism of *C. grandiflora*. The results could provide theoretical references for the exploration of a reasonable artificial cultivation mode of *C. grandiflora*, and for the enhancement of its medicinal value.

## Materials and methods

### Chemicals

Deionized water was produced by a Milli-Q water purification system (Millipore, USA). Methanol, acetonitrile of LC-MS grade were purchased from Thermo (USA). Formic acid was obtained from TCI (Shanghai, China). Ammonium formate was purchased from Sigma (USA). 2-chlorophenyllalanine with purity of 98.5% was obtained from Aladdin (Shanghai, China).

### Mixed sowing of *C. grandiflora* and possible host plants

The seeds used in the experiments were collected from Hekou County, Honghe Prefecture, Yunnan Province, and authenticated by Professor Li Yuchuan. The seeds of *C. grandiflora* and different host plants (Table 1) were mixed and sown in pots according to the ratio of 1:0.7. 10 pots for each treatment, with three replications. A blank control was set (only *C. grandiflora* seeds were sown). The seeds were scattered on the substrate which was watered thoroughly. After sowing, the pots were covered with a white film for moisturizing, and placed in the greenhouse for cultivation. The moisture preservation and ventilation were paid attention to

**Table 1. Information related to possible host plants investigated and identified.**

| Family | Name of the possible host plants |
|---|---|
| Cyperaceae | *Cyperus iria*L. |
| | *Fimbristylis dichotoma* (L.) Vahl |
| | *Cyperus rotundus* L. |
| | *Kyllinga brevifolia* Rottb |
| Gramineae | *Lophatherum gracile* Brongn. |
| | *Imperata cylindrica* (L.) P. Beauv. |
| | *Heteropogon contortus* (L.) Beauv. |
| Polygonaceae | *Polygonum viscosum* Buch.-Ham.exD.Don. |
| Scrophulariaceae | *Pterygiella bartschioides* Hand. Mazz. |

every day and the watering was carried out with a pressure sprinkler, so as not to wash away the seeds. After one month, the film and the non-host weeds were removed. The earliest emergence days and emergence rate were measured on the 60th day. The plant height, leaf numbers and survival rates were recorded on the 180th day. After 300 days of cultivation, the plant fresh weight and root fresh weight were measured, and the parasitic situations at the roots were observed. Take the average of three replicates for analysis.

## Judgement basis of parasitic relationship between *C. grandiflora* and host plants

After mixed sowing and cultivated for 300 days, the survival status, root intertwined situation, and the haustoria number of *C. grandiflora* were examined to determine whether companion plants could establish parasitic relationships with *C. grandiflora*.

## Handling of experimental materials used in nontargeted metabolomics analysis

The experiments adopted a potted planting method, with peat, vermiculite, and perlite as the sowing substrates, prepared in a 3:1:1 volume ratio and were carried out in the agronomy practice garden of Kunming University. The pots used had a diameter of 15 cm and a height of 15 cm. On January 6, 2020, the seeds of *C. grandiflora* and the host *Cyperus iria* were mixed and sown in a 1:1 ratio. Ten seeds for each pot, and a total of 20 pots were sown. All plants were cultured under natural conditions and watered promptly according to the moisture content of the substrate. The seeds of *C. grandiflora* began to germinate one after another (Fig 1A) after 15 days of *Cyperus iria* seeds' germination. When *C. grandiflora* grew into adult plants (Fig 1B), On August 30, 2020, 10 pots were randomly selected, and the host plant *Cyperus iria* were completely removed and were set as non-parasitic *C. grandiflora* (WJ) (Fig 1C), while the remaining 10 pots remained in their original state and were set as parasitic *C. grandiflora* (JS). The management methods after treatments remained unchanged for 45 days. All the *C. grandiflora* plants were collected for subsequent analysis on October 14, 2020.

## Metabolite extraction method

Each sample (200 mg) was infused with 0.6 mL 2-chlorophenylalanine methanol solution (4 mg/kg, -20°C) and mixed thoroughly by vortexing for 30s, then the sample was ground in the tissue grinder (Xinzhi, SCIENTZ-48) with 100mg glass beads at 60 Hz for 90 s. After sonicated at room temperature for 15 min, the sample solution was centrifuged at 12000 rpm and 25°C for 10 min. The supernatant was collected (300 μL) and filtered with 0.22 μm membrane (Jin

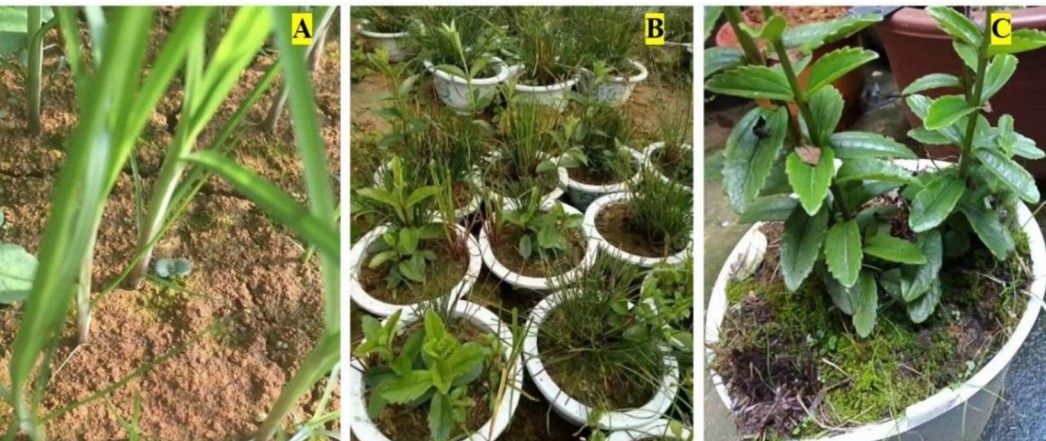

**Fig 1. Cultivation of experimental materials.**

Teng, PTFE) into the sample vial for LC-MS analysis. In addition, quality control (QC) samples were prepared by pooling the same aliquot (20 μL) of extracts from all the samples, mixed thoroughly and then transferred into a sample vial for analysis.

## Untargeted metabolomics analysis

Chromatographic separations were conducted on a Thermo Vanquish series ACQUITY UPLC system equipped with a HSS T3 column (2.1 mm × 150 mm) at 40°C. The mobile phase consisted of water with 0.1% (v/v) formic acid (phase B1) and acetonitrile with 0.1% (v/v) formic acid (phase A1) for the positive ion mode; water with 5 mM ammonium formate (phase B3) and pure acetonitrile (phase A3) for the negative ion mode at a flow rate of 0.25 mL/min. This mobile phase system was run under gradient elution as follows: 2% A1/A3 at 0–1 min; 2%-50% A1/A3 at 1–9 min; 50%-98% A1/A3 at 9–12 min; 98% A1/A3 at 12–13.5 min; 98%-2% A1/A3 at 13.5–14 min; 2% A1 at 14–20 min in the positive-ion mode (2% A3 at 14–17 min in the negative-ion mode). The auto-sampler temperature was maintained at 8°C and the injection volume was 2 μL.

The separated components were detected with a Thermo Q Exactive Plus MS spectrometer, electrospray ionization (ESI) was used for mass spectrometry detection with positive and negative ion modes. The QC samples were initially injected to equilibrate the column prior to the injections of the all samples and run (i.e. 1 injection) after every five injections of the samples. The spray voltages of positive ions and negative ions were 3.50 and 2.50 kV, respectively. Other operating parameters were as follows: sheath gas, 30 arb; auxiliary gas, 10 arb; capillary temperature, 325°C; resolution, 70000; and scan range, 81–1000 m/z. The secondary cracking was performed with HCD at a collision voltage of 30 eV, and unnecessary MS/MS information was removed by using the dynamic exclusion method.

## Metabolomics data acquisition and processing

Two treatments were set up for the experiment, namely parasitic *C. grandiflora* (JS) and non-parasitic *C. grandiflora* (WJ), with 6 replicates for each treatment. The raw data were processed using Proteowizard software for baseline filtering, peak identification, integration, retention time correction, peak alignment, and normalization, resulting in a data matrix of retention time, mass-to-charge ratio, and peak intensity. The data matrix was subjected to multivariate statistical analysis such as PCA, PLS-DA, and OPLS-DA. Finally, a combination of OPLS-DA

and t-test was used to screen for differential metabolites (VIP>1.0 and P<0.05). The differential metabolites were subjected to HMDB, LIPID MAPS and MBRole to obtain CID numbers, and then the related metabolic pathways were analyzed in KEGG based on the CID numbers.

## Results and discussion

### Growth status of *C. grandiflora* after mixed sowing with possible host plants

At present, there are few studies on the resource ecology, cultivation, biology and parasitic characteristics of *C. grandiflora*. In the early stage of our research, we found that *C. grandiflora* has a chordal fibrous root, with no distinction between main and lateral roots. The root hairs were tiny and few or severely degraded, and the roots were endowed with nodular haustoria, through which the roots of *C. grandiflora* were intertwined with that of various herbaceous plants (Fig 2). Thus, it could be assumed that the main reason why the artificial cultivation of *C. grandiflora* has not been achieved so far was that the root hemiparasitism has not been taken into account, and it has only been cultivated as a general medicinal plant. Therefore, figuring out the inherent laws of its parasitism and host selectivity is the core scientific issue to change the endangered status of *C. grandiflora* and achieve its artificially efficient cultivation.

In this study, 9 companion plant seeds were mixed sown with *C. grandiflora* seeds, and the earliest emergence days, the emergence rates, seedling growth status and yield of *C. grandiflora* at different growth stages were observed. As shown in Table 2, there were significant differences in germination and emergence time between seeds of *C. grandiflora* with different companion plants ($P<0.05$). The dominant companion plant seeds showed significant promotion effect on the growth of *C. grandiflora*. When the seeds of *C. grandiflora* were planted alone or

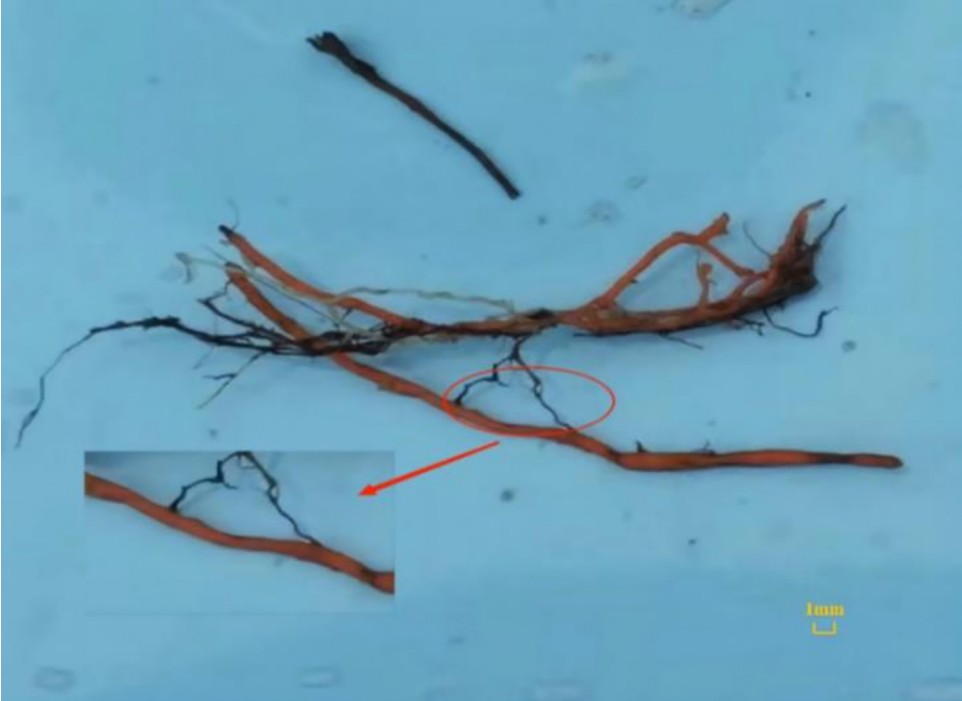

**Fig 2. Haustoria connections between *C. grandiflora* and the host (red: roots of *C. grandiflora*, black: roots of the host).**

**Table 2. Growth status of *C. grandiflora* and 9 companion plants after mixed sowing.**

| Name | Earliest emergence days/d | 60d | 180d | | | 270d | | |
|---|---|---|---|---|---|---|---|---|
| | | Emergence rate/% | Plant height/ cm | leaf numbers | Survival rate/ % | Root fresh weight/g | Plant fresh weight/g | Haustoria Number |
| *Cyperus iria* L. | 7±1a | 21.8±0.3b | 32.1±0.4a | 15.2 ± 0.2a | 97.7 ± 0.2a | 56.4 ± 0.2a | 161.1 ± 2.6a | 5.3 ± 0.1b |
| *Fimbristylis dichotoma* (L.) Vahl | 7±1a | 21.9±0.2b | 24.8 ± 0.2b | 13.6 ± 0.2b | 69.9 ± 0.2c | 30.2 ± 0.2b | 94.3 ± 1.2b | 5.0 ± 0.2c |
| *Cyperus rotundus* L. | 13±1c | 12.4±0.5c | 13.5 ± 0.2c | 9.4 ± 0.2e | 42.0 ± 0.1e | 18.5 ± 0.2d | 56.0 ± 0.5d | 6.6 ± 0.1a |
| *Kyllinga brevifolia* Rottb | 12±1c | 11.9±0.4d | 0.0 | 0.0 | 0.0 | 0.0 | 0.0 | 0.0 |
| *Lophatherum gracile* Brongn. | 7±1a | 22.4±0.4a | 13.4 ± 0.2c | 11.6 ± 0.1c | 72.6 ± 0.2b | 24.2 ± 0.2c | 69.1 ± 0.5c | 1.0 ± 0.1e |
| *Imperata cylindrica* (L.) P. Beauv. | 22±2b | 6.8±0.1f | 10.7 ± 0.1d | 11.2 ± 0.2d | 45.0 ± 0.2d | 5.4 ± 0.2e | 18.6 ± 0.6e | 1.1 ± 0.1e |
| *Heteropogon contortus* (L.) Beauv. | 22±2b | 7.6±0.2e | 11.0 ± 0.2d | 11.4 ± 0.1cd | 42.0 ± 0.3e | 4.8 ± 0.1f | 17.1 ± 0.2e | 1.6 ± 0.2d |
| *Polygonum viscosum* Buch.-Ham.exD.Don. | 25±1ab | 0.3±0.1g | 0.0 | 0.0 | 0.0 | 0.0 | 0.0 | 0.0 |
| *Pterygiella bartschioides* Hand. Mazz. | 24±2ab | 0.4±0.1g | 0.0 | 0.0 | 0.0 | 0.0 | 0.0 | 0.0 |
| *C. grandiflora* Only (CK) | 24±2a | 0.3±0.1g | 0.0 | 0.0 | 0.0 | 0.0 | 0.0 | 0.0 |

Note: Different letters indicate significant differences between data in the same column ($P<0.05$)

mixed-sowed with the seeds of 4 companion plants as *Imperata cylindrica* (L.) P. Beauv., the earliest emergence days of *C. grandiflora* seeds were around (22±2) d, and the seedling stage was delayed significantly. On the 60th day, the average emergence rate in the field was relatively low, even lower than 0.5% and seedlings that had sprouted were also poorly grown and gradually died within 180 days.

When the seeds of *C. grandiflora* were mixed-sowed with 3 companion plant seeds, namely, *Cyperus iria* L., *Fimbristylis dichotoma* (L.) Vahl and *Lophatherum gracile* Brongn., the seedlings gradually emerged from the 7th day. By the 60th day, the emergence rate could reach more than 10%. The emergence rate was significantly higher than that with non-host plants, such as *Kyllinga brevifolia* Rottb and *Heteropogon contortus* (L.) Beauv., or control *(P<0.05)*. In the mixed-sowed treatment with *Cyperus iria* L., *Fimbristylis dichotoma* (L.) Vahl and *Lophatherum gracile* Brongn., *C. grandiflora* could achieve blossoming and fruiting in the same year, completing its life cycle, and the root fresh weight could reach (56.4±0.2) g, (30.2 ±0.2) g and (24.2±0.1) g at 300th day, respectively. The roots biological yields of *C. grandiflora* were significantly increased compared with that sowed with the non-host plants *(P<0.05)*.

In terms of haustoria quality, the haustoria produced with *Cyperus iria* L. as host plants were significantly better than with other plants *(P < 0.05)*, followed by that with *Fimbristylis dichotoma* (L.) Vahl as host plants. Under the mixed-sowing treatments, the roots of the above two hosts and *C. grandiflora* were obviously entangled, and the haustoria number was 5.3 and 5.0, respectively. Comparing with companion plants from other families and genera, *Cyperus iria* L. and *Fimbristylis dichotoma* (L.) Vahl as host plants could significantly promote the seed germination, seedling growth and adult plant growth of *C. grandiflora*.

The above results indicated that *Cyperus iria* L., *Fimbristylis dichotoma* (L.) Vahl and *Lophatherum gracile* Brongn. were the dominant host plants of *C. grandiflora*., with *Cyperus iria* L. being the best host plant. Therefore, *Cyperus iria* L. was selected for the parasitic treatment in the subsequent non-targeted metabolomic study.

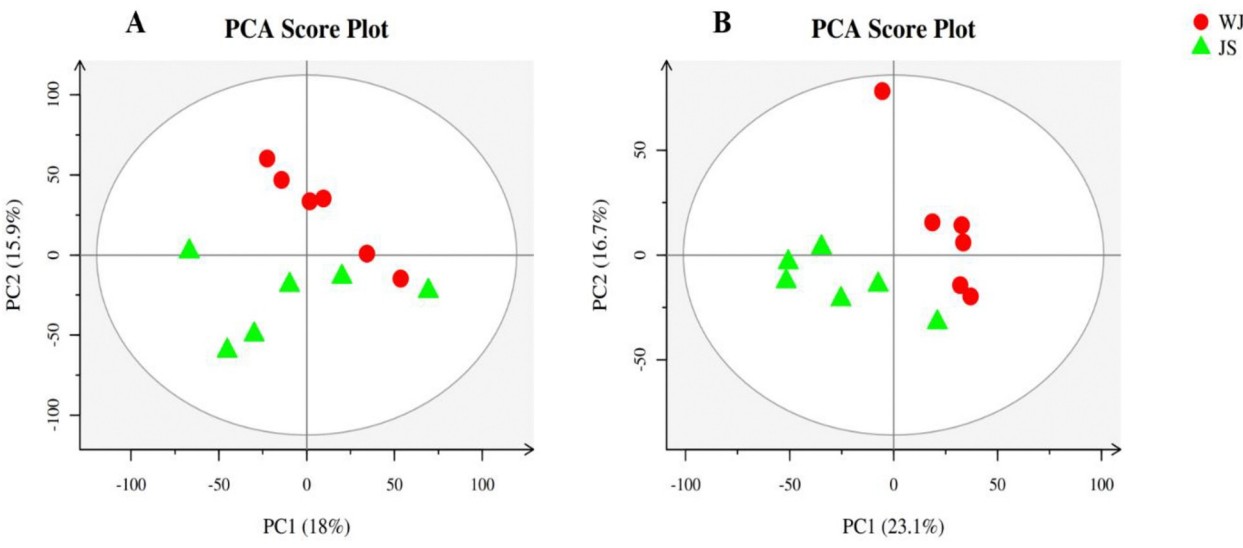

**Fig 3.** PCA analysis of metabolomics of JS and WJ samples in positive (left) and negative ion mode (right).

## Metabolomics analysis of parasitic (JS) and non-parasitic (WJ) *C. grandiflora*

**Principal component analysis of metabolites in JS and WS samples.**   The differences between sample groups and the degree of variation among samples within a group could be reflected by PCA analysis of the metabolome of quality control samples under parasitic and non-parasitic treatments [8–10]. In positive ion mode, four principal components were distinguished by PCA analysis, with a contribution rate of 18% for PC1 and 15.9% for PC2. In negative ion mode, there were three principal components, with a contribution rate of 23.1% for PC1 and 16.7% for PC2 (Fig 3). The PCA plots exhibited a clear separation trend between the two groups of samples, indicating significant differences in metabolic compounds between parasitic and non-parasitic *C. grandiflora*. The components of the parallel samples in the quality control group were close to each other and clustered near the center of the PCA plots, which indicated that there was no significant deviation during the measurement process, and that the stability of the method and metabolome data was good.

**OPLS-DA analysis of metabolites in JS and WS samples.**   The relationship model between metabolite expression and sample categories was established by OPLS-DA statistical method, and the non-orthogonal and orthogonal variables were analyzed respectively to achieve sample category prediction and metabolite intergroup difference analysis. The OPLS-DA results showed that all groups in the model were within the confidence interval, while the parameters $R^2Y$ and $Q^2$ were greater than 0.9 and 0.5, indicating that the model was reliable. As it could be seen in Fig 4, two sets of samples were distributed within the confidence interval, with complete separation and no overlapping areas, which means that there were significant differences between metabolites of parasitic (JS) and non-parasitic *C. grandiflora* (WJ). To avoid overfitting of supervised models during the modeling process, the permutation test was used to test the alignment of the OPLS-DA model to ensure the validity of the model. The $Q^2$ of the stochastic model gradually decreased with the gradual decrease of the permutation retention (Fig 5). The $Q^2$ intercepts generated by the permutation test in positive and negative ion modes were 0.2668 and 0.4265, respectively (both <0.5), indicating that there is no overfitting phenomenon in the original model, and the original model had good robustness.

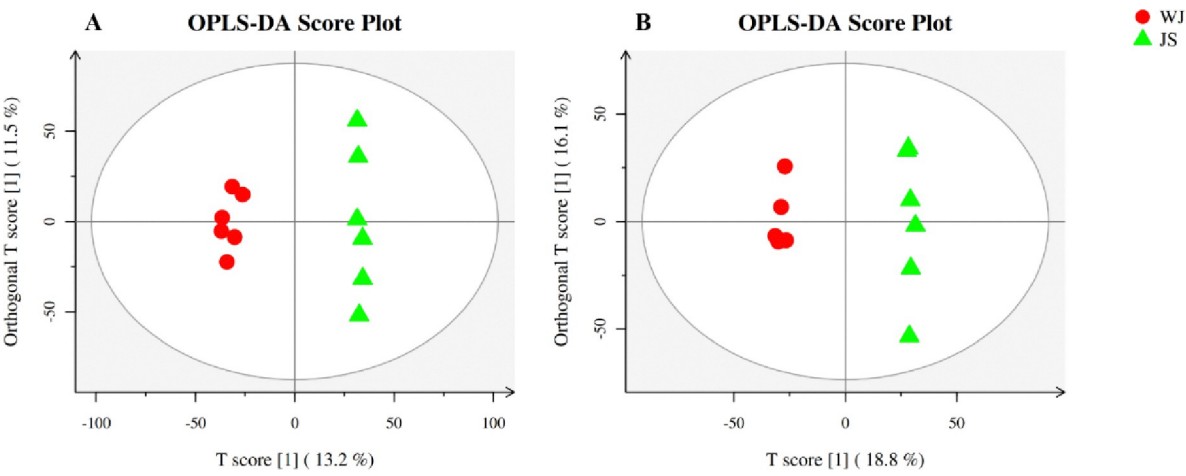

**Fig 4.** OPLS-DA analysis of metabolomics in positive (left) and negative ion modes (right) between JS and WJ samples.

**Differential metabolites screening.** Based on the OPLS-DA analysis, differential metabolites between JS and WJ samples were screened with VIP $\geq$ 1 and p-value $\leq$ 0.05 as the criteria. In the positive ion mode, 987 differential metabolites were identified, including 545 upregulated metabolites and 442 downregulated metabolites, while in the negative ion mode, 1100 differential metabolites were identified, including 545 upregulated metabolites and 555 downregulated metabolites. The differential metabolites were further identified, and a total of 82 significantly different metabolites were identified (Table 3). Cluster heatmap is a method of calculating the distance between metabolites or samples based on their expression levels [11, 12]. Similar metabolites may have similar functions and participate in the same metabolic pathway together. As shown in Fig 6, 82 significantly different metabolites of JS and WJ samples were clearly divided into two clusters, indicating significant differences in metabolites between parasitic and non-parasitic *C. grandiflora*.

Of the 82 differential metabolites, the up-regulated metabolites of the WJ and JS samples were listed in Table 3. In the 57 up-regulated metabolites of WJ samples, there were 36 compounds belong to Organic oxygen compounds, Organic acids and derivatives and Lipids and

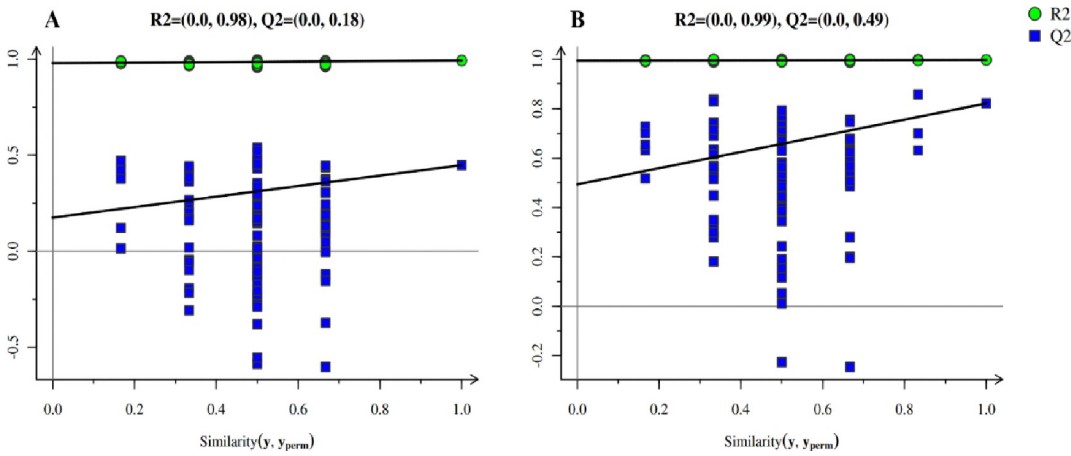

**Fig 5.** OPLS-DA permutation test of metabolomics in positive (left) and negative ion modes (right).

**Table 3. Identification information of the 82 differential metabolites between JS and WJ samples.**

| number | mode | metabolites | M/z | Mass measurement accuracy (ppm) | Retention time (min) | molecular formula | adduct ions | classification |
|---|---|---|---|---|---|---|---|---|
| Organic oxygen compounds: 13 up-regulated compounds for WJ and 4 up-regulated compounds for JS | | | | | | | | |
| 1 | ESI⁻ | D-Mannose | 180.06 | 0.30 | 89.93365 | $C_6H_{12}O_6$ | [M]⁻ | Carbohydrates and carbohydrate conjugates |
| 2 | ESI⁻ | Galactitol | 181.07 | 24.97 | 332.166 | $C_6H_{14}O_6$ | [M-H]⁻ | Carbohydrates and carbohydrate conjugates |
| 3 | ESI⁻ | L-Fucose | 163.06 | 1.96 | 525.481 | $C_6H_{12}O_5$ | [M-H]⁻ | Carbohydrates and carbohydrate conjugates |
| 4 | ESI⁻ | Gluconolactone | 177.034 | 2.71 | 92.07505 | $C_6H_{10}O_6$ | [M-H]⁻ | Carbohydrates and carbohydrate conjugates |
| 5 | ESI⁻ | L-Ribulose | 131.03 | 11.91 | 92.0949 | $C_5H_{10}O_5$ | [M-H₂O-H]⁻ | Carbohydrates and carbohydrate conjugates |
| 6 | ESI⁻ | Raffinose | 485.16 | 25.69 | 401.253 | $C_{18}H_{32}O_{16}$ | [M-H₂O-H]⁻ | Carbohydrates and carbohydrate conjugates |
| 7 | ESI⁻ | Ribonic acid | 147.03 | 6.11 | 100.387 | $C_5H_{10}O_6$ | [M-H₂O-H]⁻ | Carbohydrates and carbohydrate conjugates |
| 8 | ESI⁻ | Ribose 1,5-bisphosphate | 309.17 | 9.74 | 763.789 | $C_5H_{12}O_{11}P_2$ | [M-H]⁻ | Carbohydrates and carbohydrate conjugates |
| 9 | ESI⁻ | Sedoheptulose | 191.056 | 3.01 | 97.7987 | $C_7H_{14}O_7$ | [M-H₂O-H]⁻ | Carbohydrates and carbohydrate conjugates |
| 10 | ESI⁺ | Streptozocin | 266.10 | 5.14 | 149.9285 | $C_8H_{15}N_3O_7$ | [M+H]⁺ | Carbohydrates and carbohydrate conjugates |
| 11 | ESI⁺ | (S)-Methylmalonic acid semialdehyde | 102.03 | 4.14 | 565.574 | $C_4H_6O_3$ | [M]⁺ | Carbonyl compounds |
| 12 | ESI⁻ | 2-Heptanone | 112.98 | 1.34 | 479.368 | $C_7H_{14}O$ | [M-H]⁻ | Carbonyl compounds |
| 13 | ESI⁺ | Doxycycline | 445.16 | 4.02 | 271.2025 | $C_{22}H_{24}N_2O_8$ | [M+H]⁺ | polykeide |
| 1 | ESI⁺ | L−Iditol | 165.07 | 10.80 | 611.9925 | $C_6H_{14}O_6$ | [M + H − H₂O]⁺ | Carbohydrates and carbohydrate conjugates |
| 2 | ESI⁻ | N−Acetylmuramate | 292.10 | 3.08 | 98.90105 | $C_{11}H_{19}NO_8$ | [M−H]− | Carbohydrates and carbohydrate conjugates |
| 3 | ESI⁺ | Sorbitol | 165.07 | 13.16 | 106.094 | $C_6H_{14}O_6$ | [M + H − H₂O]⁺ | Carbohydrates and carbohydrate conjugates |
| 4 | ESI⁺ | 2,3−Butanediol | 155.11 | 1.60 | 35.07125 | $C_4H_{10}O_2S_2$ | [M + H]⁺ | Alcohols and polyols |
| Organic acids and derivatives, Organic amino compound: 13 up-regulated compounds for WJ and 7 up-regulated compounds for JS | | | | | | | | |
| 1 | ESI⁺ | (1R,6S)-6-Amino-5-oxocyclohex-2-ene-1-carboxylic acid | 156.06 | 16.87 | 263.646 | $C_7H_9NO_3$ | [M+H]⁺ | Amino acids, peptides, and analogs |
| 2 | ESI⁻ | (Z)-1-(L-Cysteinylglycine-S-yl)—N-hydroxy-2-phenylethane-1-imine | 292.07 | 28.25 | 171.519 | $C_{13}H_{17}N_3O_4S$ | [M-H₂O-H]⁻ | Amino acids, peptides, and analogs |
| 3 | ESI⁺ | 3-Methyl-L-tyrosine | 195.10 | 1.84 | 489.04 | $C_{10}H_{13}NO_3$ | [M]⁺ | Amino acids, peptides, and analogs |
| 4 | ESI⁻ | beta-Tyrosine | 164.07 | 22.52 | 287.4525 | $C_9H_{11}NO_3$ | [M+H-H₂O]⁺ | Amino acids, peptides, and analogs |
| 5 | ESI⁺ | Betaine | 118.09 | 0.03 | 91.7341 | $C_5H_{11}NO_2$ | [M+H]⁺ | Amino acids, peptides, and analogs |
| 6 | ESI⁺ | Guanidoacetic acid | 117.05 | 4.85 | 93.0056 | $C_3H_7N_3O_2$ | [M]⁺ | Amino acids, peptides, and analogs |
| 7 | ESI⁺ | L-Dopa | 198.08 | 3.20 | 124.067 | $C_9H_{11}NO_4$ | [M+H]⁺ | Amino acids, peptides, and analogs |
| 8 | ESI⁺ | L-Methionine | 150.06 | 25.54 | 103.623 | $C_5H_{11}NO_2S$ | [M+H]⁺ | Amino acids, peptides, and analogs |
| 9 | ESI⁺ | L-Proline | 116.07 | 0.86 | 97.5858 | $C_5H_9NO_2$ | [M+H]⁺ | Amino acids, peptides, and analogs |
| 10 | ESI⁺ | Tetrahydrodipicolinate | 172.06 | 15.90 | 176.9115 | $C_7H_9NO_4$ | [M+H]⁺ | Amino acids, peptides, and analogs |
| 11 | ESI⁻ | Fumaric acid | 114.99 | 12.80 | 106.484 | $C_4H_4O_4$ | [M-H]⁻ | Carboxylic acids and derivatives |
| 12 | ESI- | Hydroxypropionic acid | 89.02 | 22.96 | 205.849 | $C_3H_6O_3$ | [M-H]⁻ | Hydroxy acids and derivatives |
| 13 | ESI+ | Ureidopropionic acid | 133.06 | 29.16 | 187.092 | $C_4H_8N_2O_3$ | [M+H]⁺ | Organic carbonic acids and derivatives |
| 1 | ESI⁻ | beta−Alanine | 89.02 | 5.43 | 574.923 | $C_3H_7NO_2$ | [M]− | Amino acids, peptides, and analogs |
| 2 | ESI⁻ | beta−Citryl−L−glutamate | 320.06 | 2.24 | 331.176 | $C_{11}H_{15}NO_{10}$ | [M−H]− | Amino acids, peptides, and analogs |
| 3 | ESI⁻ | gamma−Aminobutyric acid | 102.96 | 1.41 | 42.4916 | $C_4H_9NO_2$ | [M]− | Amino acids, peptides, and analogs |
| 4 | ESI⁻ | L−Alanine | 89.02 | 7.61 | 395.904 | $C_3H_7NO_2$ | [M]− | Amino acids, peptides, and analogs |
| 5 | ESI⁻ | Citric acid | 191.02 | 1.55 | 73.1904 | $C_6H_8O_7$ | [M−H]− | Carboxylic acids derivatives |
| 6 | ESI− | Isocitric acid | 191.02 | 5.04 | 102.642 | $C_6H_8O_7$ | [M−H]− | Carboxylic acids derivatives |
| 7 | ESI+ | Ciliatine | 125.02 | 2.61 | 366.628 | $C_2H_8NO_3P$ | [M]+ | Organic phosphoric acids and derivatives |

*(Continued)*

**Table 3.** (Continued)

| number | mode | metabolites | M/z | Mass measurement accuracy (ppm) | Retention time (min) | molecular formula | adduct ions | classification |
|---|---|---|---|---|---|---|---|---|
| Lipids and lipid-like molecules: 9 up-regulated compounds for WJ and 8 up-regulated compounds for JS | | | | | | | | |
| 1 | ESI- | 13(S)-HPOT | 309.20 | 5.65 | 768.9055 | $C_{18}H_{30}O_4$ | [M-H]- | Fatty Acyls |
| 2 | ESI- | 9-OxoODE | 293.21 | 1.91 | 768.372 | $C_{18}H_{30}O_3$ | [M-H]- | Fatty Acyls |
| 3 | ESI+ | Dethiobiotin | 215.14 | 2.18 | 145.134 | $C_{10}H_{18}N_2O_3$ | [M+H]+ | Fatty Acyls |
| 4 | ESI- | Methyl jasmonate | 223.13 | 2.42 | 732.66 | $C_{13}H_{20}O_3$ | [M-H]- | Fatty Acyls |
| 5 | ESI- | Maslinic acid | 471.34 | 6.02 | 826.8215 | $C_{30}H_{48}O_4$ | [M-H]- | Prenol lipids |
| 6 | ESI- | Ursolic acid | 455.35 | 3.02 | 735.7375 | $C_{30}H_{48}O_3$ | [M-H]- | Prenol lipids |
| 7 | ESI- | 25-Hydroxycholesterol | 401.09 | 1.55 | 505.27 | $C_{27}H_{46}O_2$ | [M-H]- | Steroids and steroid derivatives |
| 8 | ESI- | Cortisol | 361.16 | 2.18 | 648.395 | $C_{21}H_{30}O_5$ | [M-H]- | Steroids and steroid derivatives |
| 9 | ESI+ | Wortmannin | 429.16 | 21.83 | 327.433 | $C_{23}H_{24}O_8$ | [M+H]+ | Steroids and steroid derivatives |
| 1 | ESI+ | 12−Keto−tetrahydro−leukotriene B4 | 336.22 | 18.77 | 763.7655 | $C_{20}H_{32}O_4$ | [M]+ | Fatty Acyls |
| 2 | ESI+ | 16(R)−HETE | 303.23 | 4.31 | 719.212 | $C_{20}H_{32}O_3$ | $[M + H − H_2O]+$ | Fatty Acyls |
| 3 | ESI- | 20−Hydroxy−leukotriene B4 | 333.21 | 1.03 | 788.114 | $C_{20}H_{32}O_5$ | $[M − H_2O − H]−$ | Fatty Acyls |
| 4 | ESI+ | EPA (d5) | 285.22 | 10.92 | 708.833 | $C_{20}H_{30}O_2$ | $[M + H − H_2O]+$ | Fatty Acyls |
| 5 | ESI+ | Oleic acid | 283.26 | 6.06 | 64.8969 | $C_{18}H_{34}O_2$ | [M + H]+ | Fatty Acyls |
| 6 | ESI+ | Valeric acid | 103.08 | 0.90 | 33.19865 | $C_5H_{10}O_2$ | [M + H]+ | Fatty Acyls |
| 7 | ESI+ | Capsidiol | 219.17 | 0.46 | 524.378 | $C_{15}H_{24}O_2$ | $[M + H − H_2O]+$ | Prenol lipids |
| 8 | ESI+ | T2 Toxin | 467.23 | 1.81 | 524.586 | $C_{24}H_{34}O_9$ | [M + H]+ | Prenol lipids |
| Benzenoids: 10 up-regulated compounds for WJ and 1 up-regulated compounds for JS | | | | | | | | |
| 1 | ESI+ | 3,4-Dihydroxymandelic acid | 185.04 | 15.40 | 246.22 | $C_8H_8O_5$ | [M+H]+ | Phenols Phenols |
| 2 | ESI- | 4-Nitrophenol | 138.02 | 10.87 | 568.525 | $C_6H_5NO_3$ | [M-H]- | Phenols Phenols |
| 3 | ESI+ | Aminohydroquinone | 126.05 | 5.60 | 254.374 | $C_6H_7NO_2$ | [M+H]+ | Phenols Phenols |
| 4 | ESI+ | Phenylephrine | 150.09 | 25.84 | 103.954 | $C_9H_{13}NO_2$ | $[M+H-H_2O]+$ | Phenols Phenols |
| 5 | ESI+ | 2-Aminobenzoic acid | 138.05 | 2.63 | 97.6882 | $C_7H_7NO_2$ | [M+H]+ | Benzene and substituted derivatives |
| 6 | ESI- | 4-(beta-D-Glucosyloxy) benzoate | 299.08 | 1.54 | 200.035 | $C_{13}H_{16}O_8$ | [M-H]- | Benzene and substituted derivatives |
| 7 | ESI- | Benzoic acid | 121.03 | 12.59 | 139.0705 | $C_7H_6O_2$ | [M-H]- | Benzene and substituted derivatives |
| 8 | ESI+ | procaine | 236.15 | 18.76 | 102.6365 | $C_{13}H_{20}N_2O_2$ | [M]+ | Benzene and substituted derivatives |
| 9 | ESI- | Phenylethylamine | 121.03 | 1.60 | 223.476 | $C_8H_{11}N$ | [M]- | Benzene and substituted derivatives |
| 10 | ESI- | Phenylpyruvic acid | 164.04 | 1.71 | 272.407 | $C_9H_8O_3$ | [M]- | Benzene and substituted derivatives |
| 1 | ESI+ | 1,2,3−Trihydroxybenzene | 127.04 | 0.031 | 286.1795 | $C_6H_6O_3$ | [M + H]+ | Phenols Phenols |
| Organic heterocyclic compounds: 9 up-regulated compounds for WJ and 4 up-regulated compounds for JS | | | | | | | | |
| 1 | ESI+ | 4,5,6,7-Tetrahydroisoxazolo(5,4-c)pyridin-3-ol | 123.05 | 14.13 | 144.0275 | $C_6H_8N_2O_2$ | $[M+H-H_2O]+$ | Azoles |
| 2 | ESI- | Dihydrouracil | 112.98 | 6.95 | 379.095 | $C_4H_6N_2O_2$ | [M-H]- | pyrimidines |
| 3 | ESI+ | Nalidixic Acid | 233.09 | 2.83 | 389.514 | $C_{12}H_{12}N_2O_3$ | [M+H]+ | Naphthyridines |
| 4 | ESI+ | Oxolinic acid | 262.07 | 2.65 | 289.064 | $C_{13}H_{11}NO_5$ | [M+H]+ | Quinolines and derivatives |
| 5 | ESI+ | N-Glucosylnicotinate | 287.09 | 19.52 | 97.5896 | $C_{12}H_{16}NO_7$ | [M+H]+ | Pyridine and derivatives |
| 6 | ESI+ | Nicotinic acid | 124.04 | 1.62 | 102.666 | $C_6H_5NO_2$ | [M+H]+ | Pyridine and derivatives |

(*Continued*)

**Table 3.** (Continued)

| number | mode | metabolites | M/z | Mass measurement accuracy (ppm) | Retention time (min) | molecular formula | adduct ions | classification |
|---|---|---|---|---|---|---|---|---|
| 7 | ESI⁻ | Indole-3-acetyl-myo-inositol | 337.12 | 15.82 | 328.5205 | $C_{16}H_{19}NO_7$ | [M]- | Indoles and derivatives |
| 8 | ESI⁺ | L-Tryptophan | 204.09 | 18.60 | 94.1302 | $C_{11}H_{12}N_2O_2$ | [M]+ | Indoles and derivatives |
| 9 | ESI⁺ | Cyclopeptine | 280.12 | 15.68 | 249.7065 | $C_{17}H_{16}N_2O_2$ | [M]+ | Benzodiazepine |
| 1 | ESI⁻ | 10–Formyldihydrofolate | 452.14 | 15.09 | 826.194 | $C_{20}H_{21}N_7O_7$ | [M–H₂O–H]– | Pterins and derivatives |
| 2 | ESI⁻ | Theobromine | 179.05 | 14.38 | 457.202 | $C_7H_8N_4O_2$ | [M–H]– | Purines Purines |
| 3 | ESI⁺ | GF 109203X | 413.21 | 27.94 | 458.0895 | $C_{25}H_{24}N_4O_2$ | [M + H]+ | Indoles and derivatives |
| 4 | ESI⁺ | Myristicin | 193.09 | 1.15 | 666.929 | $C_{11}H_{12}O_3$ | [M + H]+ | Benzodioxoles |
| Nucleosides, nucleotides, and analogs: 3 up-regulated compounds for WJ | | | | | | | | |
| 1 | ESI⁻ | Thymidine thymine | 223.03 | 4.03 | 723.8955 | $C_{10}H_{14}N_2O_5$ | [M-H₂O-H]- | Pyrimidine nucleosides |
| 2 | ESI⁻ | Uridine diphosphate glucose | 565.04 | 10.63 | 81.9273 | $C_{15}H_{24}N_2O_{17}P_2$ | [M-H]- | Pyrimidine nucleotides |
| 3 | ESI⁺ | S-Adenosylmethionine | 398.24 | 4.83 | 761.1315 | $C_{15}H_{22}N_6O_5S$ | [M]+ | 5'-deoxyribonucleosides |
| Phenylpropanoids and polyketones: 1 up-regulated compounds for JS | | | | | | | | |
| 1 | ESI⁻ | 4–Methylumbelliferyl acetate | 217.05 | 14.22 | 832.26 | $C_{12}H_{10}O_4$ | [M–H]– | Coumarins and derivatives |

Note: up-regulated compounds for JS were marked blue

lipid-like molecules, accounting for 63.2% of the upregulated metabolites. Organic oxygen compounds, especially carbohydrates, are essential metabolic resources for plants to build carbon skeleton and obtain energy [13, 14], while amino acids play an indispensable role in plant primary metabolism, which is important for the physiological process of plants [15]; whereas lipids are the main components of cell membranes, and play multiple roles in cellular functions such as signal transduction, membrane transport, and cytoskeleton rearrangement, and at the same time are also important precursors for terpenoid biosynthesis [16]. The main differential metabolites of non-parasitic *C. grandiflora* (WJ) samples were mostly compounds used for plant growth, which might be due to the lack of energy and compounds that non-parasitic *C. grandiflora* relies on for survival. In addition, it is worth noting that 10 kinds of benzenoids were also up-regulated in WJ samples. Benzenoids, which are usually volatile, could promote the interactions between plants and the environment, attract pollinators and seed dispersers, and protect the plants from pathogens, parasites, and herbivores [17]. In conclusion, the up-regulated compounds of WJ samples were mainly used for plants to maintain growth and pollination, and protect themselves from harm. It could be inferred that parasitic *C. grandiflora* (JS) could obtain the above-mentioned compounds through other ways. It has been reported that parasitic plants could form "physiological bridges" with the vascular tissues of the host through haustoria, so as to exchange metabolites such as proteins continuously for their own growth, blossoming and fruiting [18, 19].

Among the 25 up-regulated metabolites of JS samples, L-Alanine and Citric acid from Organic acids and derivatives are mostly related to the pathways that provide energy and nutrients such as carbon fixation and tricarboxylic acid cycle. This explained the need for WJ samples to up-regulate other metabolites in order to maintain normal growth. In addition, 4-Methylumbelliferyl acetate from Phenylpropanoids and polyketides, theobromine and myristicin from Organic heterocyclic compounds, as well as EPA (d5) from Lipids and lipid-like molecules are all metabolites with certain functional properties. Therefore, it could be inferred that non-parasitic *C. grandiflora* (WJ) samples might sacrifice the production of some functional metabolites to maintain normal growth, which ultimately affects the efficacy of non-parasitic *C. grandiflora*.

It has been reported that the main efficacy components of *C. grandiflora* are iridoid compounds, phenylethanol glycosides and azafrin, etc [20–22]. Liao et al. [23] had successfully isolated 9 Iridoid glycosides, namely, aucubin, mussaenoside, 8-epiloganin, 8-epiloganic acid, mussaenosidic acid, catalpol, gardoside methyl ester, geniposidic acid, and 6-O-methylaucubin, from the ethanol extracts of *C. grandiflora* for the first time by silica-gel column chromatography method. However, the iridoid compounds were not detected in the differential metabolites, which might be due to the fact that the main efficacy components of *C. grandiflora* require a certain period of time to accumulate, and accumulation time of the samples used in the study were insufficient, besides, the assay used could only detect compounds in the database, while the iridoid compounds are not in the database.

## KEGG functional annotation and classification of differential metabolites

Differential metabolites detected in WJ and JS samples were annotated and classified by pathway enrichment using the KEGG database. As shown in Table 4, 32 out of 82 differential metabolites were annotated into 37 KEGG pathways (Fig 7).

The biosynthetic pathways of the main efficacy components needs to be clarified for artificial cultivation of *C. grandiflora* with higher medicinal value. Iridoid glycosides belong to monoterpenoids, and their biosynthesis in plants could be divided into three stages. The first stage is precursor formation, which includes the plastidial 2-C-methyl-D-erythritol-

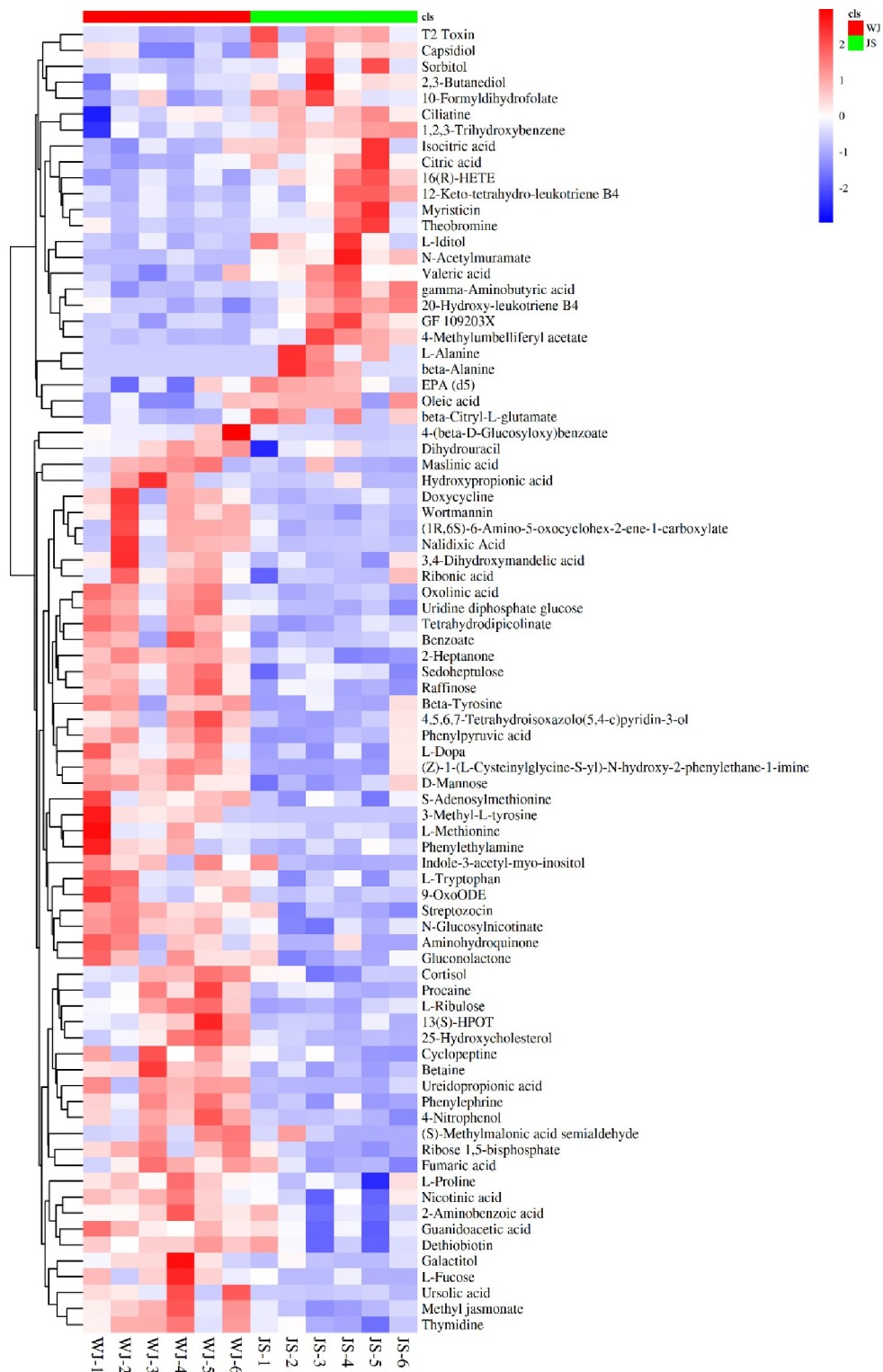

**Fig 6. Cluster analysis of metabolites in JS and WJ samples.**

**Table 4. The 37 metabolic pathways analysis of JS and WJ samples.**

| Pathway name | KEGG Pathway ID | Match status | Metabolic pathway impact value | -log (p) | Differential metabolites in target metabolic pathways |
|---|---|---|---|---|---|
| beta-Alanine metabolism | ath00410 | 3/12 | 0.75 | 3.32 | Beta Alanine (C00099); Uridopropionic acid (C02642); Dihydrouracil (C00429) |
| Pantothenate and CoA biosynthesis | ath00770 | 3/14 | 0.35 | 2.91 | Uridopropionic acid (C02642); Dihydrouracil (C00429); Beta Alanine (C00099) |
| Galactose metabolism | ath00052 | 4/26 | 0.07 | 2.55 | Raffinose (C00492); Uridine diphosphate glucose (C00029); D-Manose (C00159); Sorbitol (C00794) |
| Phenylalanine metabolism | ath00360 | 2/8 | 0.33 | 2.44 | Phenylethylamine(C05332); Phenylpyruvicpentara acid(C00166) |
| Citrate cycle (TCA cycle) | ath00020 | 3/20 | 0.18 | 2.04 | Iscitric acid (C00311); Citric acid (C00158); Functional acid (C00122) |
| Phenylalanine, tyrosine and tryptophan biosynthesis | ath00400 | 3/21 | 0.11 | 1.93 | Phenolpyruvic acid (C00166); L-Tryptophan (C00078); 2-Aminobenzoic acid (C00108) |
| Alanine, aspartate and glutamate metabolism | ath00250 | 3/22 | 0.01 | 1.83 | L-Alanine (C00041); Fuzzy acid (C00122); Gamma Aminobutyric acid (C00334) |
| Pentose and glucuronate interconversions | ath00040 | 2/12 | 0.30 | 1.74 | L-Ribulose (C00310); Uridine diphosphate glucose (C00029) |
| Arginine and proline metabolism | ath00330 | 4/38 | 0.09 | 1.52 | L-Proline (C00148); S-Adenosylmethionine (C00019); Fuzzy acid (C00122); Gamma Aminobutyric acid (C00334) |
| Pyrimidine metabolism | ath00240 | 4/38 | 0 | 1.52 | Dihydrouracil(C00429); Ureidopropionic acid(C02642);Thymidine (C00214); beta-Alanine(C00099) |
| Fructose and mannose metabolism | ath00051 | 2/16 | 0 | 1.31 | Sorbitol(C00794); D-Mannose(C00159) |
| Glyoxylate and dicarboxylate metabolism | ath00630 | 2/17 | 0.31 | 1.22 | Isocitric acid(C00311); Citric acid(C00158) |
| Tyrosine metabolism | ath00350 | 2/18 | 0 | 1.14 | L-Dopa(C00355); Fumaric acid(C00122) |
| Isoquinoline alkaloid biosynthesis | ath00950 | 1/6 | 0 | 1.12 | L-Dopa(C00355) |
| Indole alkaloid biosynthesis | ath00901 | 1/7 | 0 | 0.10 | L-Tryptophan(C00078) |
| Arachidonic acid metabolism | ath00590 | 1/8 | 0 | 0.89 | Leukotriene B4(C02165) |
| alpha-Linolenic acid metabolism | ath00592 | 2/23 | 0.23 | 0.83 | 13(S)-HPOT(C04785); Methyl jasmonate(C11512) |
| Lysine biosynthesis | ath00300 | 1/10 | 0.19 | 0.73 | Tetrahydrodipicolinate(C03972) |
| Caffeine metabolism | ath00232 | 1/10 | 0 | 0.73 | Theobromine(C07480) |
| Nicotinate and nicotinamide metabolism | ath00760 | 1/12 | 0 | 0.60 | Nicotinic acid(C00253) |
| Glycerolipid metabolism | ath00561 | 1/13 | 0 | 0.55 | Uridine diphosphate glucose(C00029) |
| Glycine, serine and threonine metabolism | ath00260 | 2/30 | 0 | 0.54 | Betaine(C00719); L-Tryptophan(C00078) |
| Ascorbate and aldarate metabolism | ath00053 | 1/15 | 0 | 0.46 | Uridine diphosphate glucose(C00029) |
| Propanoate metabolism | ath00640 | 1/15 | 0 | 0.46 | (S)-Methylmalonic acid semialdehyde(C06002) |
| Aminoacyl-tRNA biosynthesis | ath00970 | 4/67 | 0 | 0.46 | L-Methionine (C00073); L-Alanine (C00041); L-Tryptopha (C00078); L-Proline (C00148) |
| Cysteine and methionine metabolism | ath00270 | 2/34 | 0.38 | 0.43 | S-Adenosylmethionine(C00019); L-Methionine(C00073) |
| Butanoate metabolism | ath00650 | 1/18 | 0 | 0.36 | gamma-Aminobutyric acid(C00334) |
| Zeatin biosynthesis | ath00908 | 1/19 | 0 | 0.33 | Uridine diphosphate glucose(C00029) |
| Selenoamino acid metabolism | ath00450 | 1/19 | 0 | 0.33 | L-Alanine(C00041) |
| Amino sugar and nucleotide sugar metabolism | ath00520 | 2/41 | 0.15 | 0.28 | Uridine diphosphate glucose(C00029); D-Mannose(C00159) |
| Carbon fixation in photosynthetic organisms | ath00710 | 1/21 | 0 | 0.29 | L-Alanine(C00041) |
| Biosynthesis of unsaturated fatty acids | ath01040 | 2/42 | 0 | 0.27 | Oleic acid(C00712); EPA (d5)(C06428) |

(*Continued*)

**Table 4.** (Continued)

| Pathway name | KEGG Pathway ID | Match status | Metabolic pathway impact value | -log (p) | Differential metabolites in target metabolic pathways |
|---|---|---|---|---|---|
| Tryptophan metabolism | ath00380 | 1/27 | 0.17 | 0.18 | L-Tryptophan(C00078) |
| Starch and sucrose metabolism | ath00500 | 1/30 | 0.17 | 0.15 | Uridine diphosphate glucose(C00029) |
| Glucosinolate biosynthesis | ath00966 | 2/54 | 0 | 0.14 | L-Methionine(C00073); L-Tryptophan(C00078) |
| Valine, leucine and isoleucine degradation | ath00280 | 1/34 | 0.01 | 0.11 | (S)-Methylmalonic acid semialdehyde(C06002) |
| Fatty acid biosynthesis | ath00061 | 1/49 | 0 | 0.04 | Oleic acid(C00712) |

Note: Match status indicates metabolites involvement in the pathway, with the number of differential metabolites before the slash and the total number of metabolites in the pathway after the slash. The pathways were sorted by -log(P) values.

4-phosphate (MEP) pathway and the cytoplasmic mevalonate (MVA) pathway producing isopentenyl diphosphate (IPP) and dimethylallyl diphosphate (DMAPP) [24]. The second stage is the formation of a carbon skeleton structure, beginning with geranylgeranyl diphosphate and ending at the synthesis of secologanin [25, 26]. The third stage is the post- structural modification of terpenoids: hydroxylation, methylation, isomerization, demethylation, glycosylation, etc [27].

Analysis of the 37 KEGG pathways revealed that 19 pathways could be involved in the synthesis pathways of the main efficacy components of *C. grandiflora*, such as Butanoate metabolism, Glycerolipid metabolism, Pantothenate and CoA biosynthesis, Fructose and mannose metabolism, etc. The metabolites from the 19 pathways could enter Glycolysis/

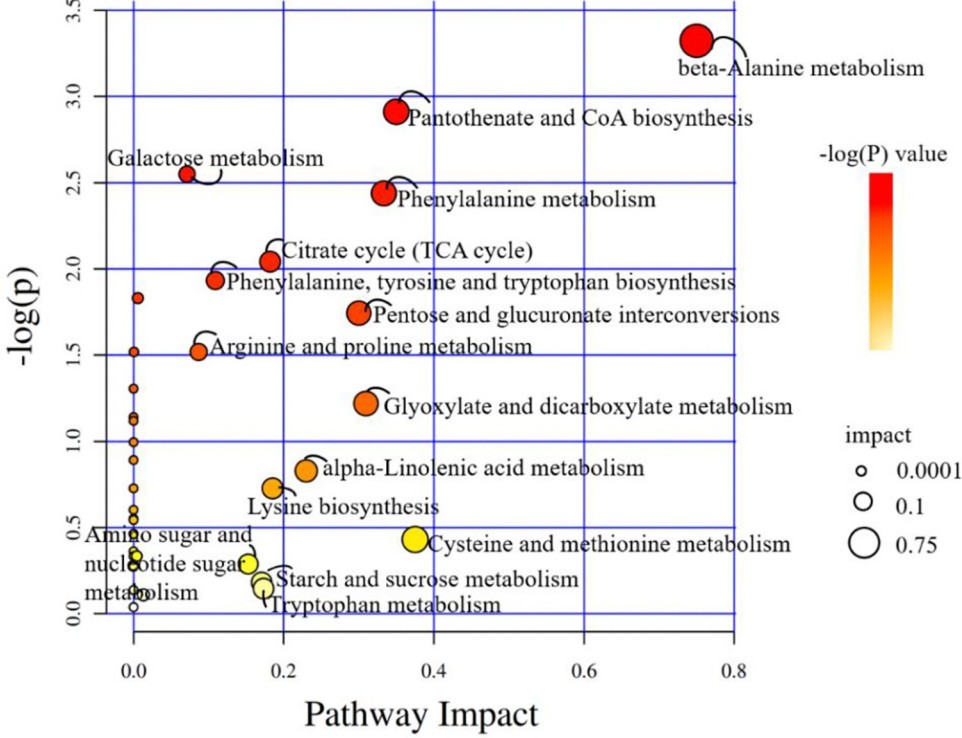

**Fig 7. Bubble plots of the metabolic pathway analysis.** The main metabolic pathways were labeled in the figure according to the pathway impact and -log(P) values.

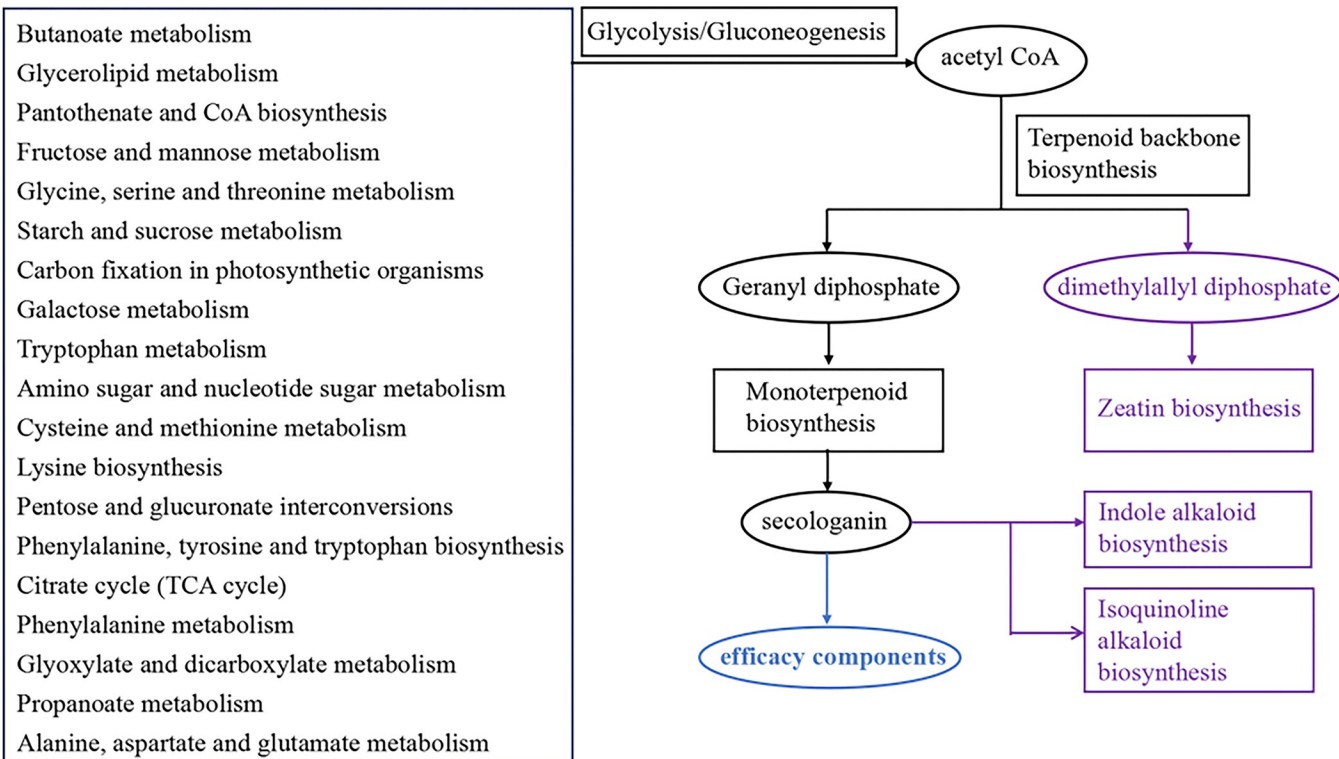

**Fig 8. The pathways for the main efficacy components of *C. grandiflora* and their possible competitive pathways (purple).**

Gluconeogenesis pathway (KEGG ID: map00010), through which acetyl CoA was generated to enter the Terpenoid backbone biosynthesis pathway (KEGG ID: map00900). In Terpenoid backbone biosynthesis pathway, geranyl diphosphate was further synthesized by various enzymes, and then entered the biosynthesis pathways of monoterpenoid. In addition to directly entering into the synthesis pathway of terpenes, acetyl CoA could also be metabolized to dimethylallyl diphosphate which could enter into the Zeatin biosynthesis pathway (KEGG ID: map00908). The metabolic pathways following the terpenoid synthesis pathway were also analyzed, the results showed that secologanin generated from the Monoterpenoid biosynthesis pathway (KEGG ID: map00902) could be further metabolized by Indole alkaloid biosynthesis (KEGG ID: map00901) and Isoquinoline alkaloid biosynthesis (KEGG ID: map00950) pathways (Fig 8).

The differential metabolites involved in metabolism were analyzed in conjunction with the metabolic pathways. The results showed that the up-regulated metabolites (e.g., L-Alanine, fumaric acid, gamma-aminobutyric acid, isocitric acid, and citric acid, etc.) of JS samples were mainly involved in the biosynthesis pathways of the main efficacy components of *C. grandiflora* (Alanine, aspartate and glutamate metabolism, Butanoate metabolism, and Citrate cycle (TCA cycle)). Multiple up-regulated differential metabolites (e.g. L-Ribulose, Uridine diphosphate glucose, Phenylethylamine, Phenylpyruvicpentara acid, Phenylpyruvic acid, L-Tryptophan and 2-Aminobenzoic acid, etc.) of WJ samples were also involved in the biosynthesis pathways of the main efficacy components. However, it was worth noting that there were also some metabolites (Uridine diphosphate glucose) of WJ samples were involved in the Zeatin biosynthesis pathway (KEGG ID: map00908) which is a competitive pathway with Monoterpenoid biosynthesis pathway (KEGG ID: map00902). Besides, some up-regulated metabolites,

such as L-Tryptophan and L-Dopa, from WJ samples were further entered the subsequent metabolic pathways of Monoterpenoid biosynthesis pathway, that is, Indole alkaloid biosynthesis and Isoquinoline alkaloid biosynthesis pathway, which might lead to might lead to a decrease in the efficacy components as iridoid compounds in the non-parasitic *C. grandiflora*.

In addition, the up-regulated metabolites of JS samples were also involved in the metabolic pathways of other functional active components, such as Arachidonic acid metabolism for EPA, Caffeine metabolism pathway for Theobromine, etc. However, the up-regulated metabolites of WJ samples were mostly involved in metabolic pathways related to carbohydrate decomposition, amino acid synthesis and utilization, energy metabolism and growth regulator synthesis. For example, Nicotinic acid was involved in Nicotinate and nicotinamide metabolism pathway, which is related to energy metabolism, with NAD being the main synthesis compound. Alpha-linolenic acid metabolism pathway involving 13(S)-HPOT and methyl jasmonate was the main pathway for jasmonic acid synthesis. Jasmonic acid (JA) and its derivatives are lipid-derived phytohormones that control plant defense responses and development, such as seed germination, root growth, tendril curling and trichome initiation [28]. Therefore, it could be inferred that the metabolic pathways involved by up-regulated metabolites of WJ samples were mainly the nutrients synthesis and catabolism, energy generation and phytohormone production for the growth needs of WJ samples, while JS samples could obtain those components from host-plants and thus up-regulate the bioactive compounds that could further enhance the nutritional value of *C. grandiflora* during the growth progress.

## Conclusions

*Centranthera grandiflora* Benth. is a kind of "food and medicine" plant in Yunnan, and has become a popular ethno-medicinal herb to be developed in Yunnan due to its ability for treating early leukemia, menopausal syndromes and other diseases without toxic side effects. However, due to issues such as excessive exploitation and environmental damage, the depletion of *C. grandiflora* resources has attracted great attentions from researchers. Currently, both artificial scale cultivation and wild cultivation of *C. grandiflora* have ended in failure, which was mainly due to insufficient understanding of its growth and cultivation characteristics.

Through the investigation of wild habitat and companion plants, our research group found that there were "haustoria" on the roots of *C. grandiflora* which are specific functional organs of parasitic plants, and thus it was assumed that *C. grandiflora* should be a root parasitic plant. In this study, nine kinds of possible host-plant seeds were mixed-sowed with the seeds of *C. grandiflora*. The results confirmed that *C. grandiflora* is a specialized root hemiparasitic plant, which could not survive and grow independently for a long period of time without a suitable host plant. *C. grandiflora* has a relatively narrow host range, and there is a clear preference for its host selection. According to the indicators acquired from the mixed sowing experiments, it was found that *Cyperus iria* L. was the best host for *C. grandiflora*.

The growth and metabolism of parasitic (JS, with *Cyperus iria* L. as host plant) and non-parasitic *C. grandiflora* (WJ) samples were analyzed by non-targeted metabolomics, and a total of 82 significant differential metabolites of JS and WJ samples were screened, of which 32 metabolites were annotated to 37 KEGG pathways. In terms of the differential metabolites of JS and WJ samples, the main up-regulated differential metabolites of WJ samples were concentrated on compounds used for plant growth while that of JS samples were on functional active compounds as EPA. As for the metabolic pathways, the up-regulated metabolites of JS samples were involved in the biosynthesis of a wide range of functional active components in addition to the biosynthesis pathways of efficacy components, while that of WJ samples were not only involved in the biosynthesis pathways of efficacy components, but also were involved in the

competitive pathways of efficacy components synthesis pathways as well as multiple metabolic pathways for the nutrients synthesis and catabolism, energy generation and phytohormone production for the growth needs of WJ samples.

In conclusion, non-parasitic *C. grandiflora* could not complete its life cycle and would gradually die with 180 days while parasitic *C. grandiflora* could achieve blossoming and fruiting in the same year, completing its life cycle. The results were mainly due to the fact that non-parasitic *C. grandiflora* could not produce enough sufficient nutrients, energy and phytohormone on its own, thus, it could only mobilize all kinds of substances to produce compounds required for its own growth through metabolic pathways, while parasitic *C. grandiflora* could obtain the compounds needed for its own growth from the host plants, sparing more compounds to synthesize efficacy components and other functional compounds. This study provides a new approach for the artificial cultivation as well as a theoretical basis the possible mechanism of root hemiparasitism and the medicinal efficacy enhancement of *C. grandiflora* through the analysis of different metabolites and metabolic pathways between parasitic and non-parasitic *C. grandiflora* samples.

Part of the primary and secondary analysis data and figures of non-targeted metabolic analysis were presented in the supporting files (S1 File) and S1–S9 Figs.

## Supporting information

**S1 Fig. Heatmap of overall metabolites in positive ion mode.**
(DOCX)

**S2 Fig. Heatmap of differential metabolites of WJ and JS samples in positive ion mode.**
(DOCX)

**S3 Fig. Heatmap of overall metabolites in negative ion mode.**
(DOCX)

**S4 Fig. Heatmap of differential metabolites of WJ and JS samples in positive ion mode.**
(DOCX)

**S5 Fig. Dendrogram of the overall samples in positive ion mode.**
(DOCX)

**S6 Fig. Dendrogram of the overall samples in negative ion mode.**
(DOCX)

**S7 Fig. Total differential metabolites in positive ion mode.**
(DOCX)

**S8 Fig. Total differential metabolites in negative ion mode.**
(DOCX)

**S9 Fig. Correlation heatmap of differential metabolites of WJ and JS samples.**
(DOCX)

**S1 File.**
(RAR)

## Author Contributions

**Data curation:** Song Jin, Yuchuan Li, Jun Ni, Haili Xie, Falin Lei.

**Formal analysis:** Jun Ni, Falin Lei.

**Funding acquisition:** Yuchuan Li, He Liu.

**Investigation:** Falin Lei.

**Methodology:** Song Jin, Haili Xie, Falin Lei.

**Software:** Haili Xie.

**Supervision:** Yuchuan Li, He Liu.

**Writing – original draft:** Song Jin, Jun Ni, Haili Xie, Falin Lei.

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
