## [Decision Letter · Decision Letter 0]

24 Jun 2024

PONE-D-24-19588Title. Host plants selection of Centranthera grandiflora Benth. and nontargeted metabolomics analysis of its parasitic and non-parasitic samplesPLOS ONE

Dear Dr. Liu,

Thank you for submitting your manuscript to PLOS ONE. After careful consideration, we feel that it has merit but does not fully meet PLOS ONE’s publication criteria as it currently stands. Therefore, we invite you to submit a revised version of the manuscript that addresses the points raised during the review process.

We look forward to receiving your revised manuscript.

Kind regards,

Rajib Chowdhury, M.Sc.; MPH

Academic Editor

PLOS ONE

Journal Requirements:

2. Please amend the title to remove the date: "Title. Host plants selection of Centranthera grandiflora Benth. and nontargeted metabolomics analysis of its parasitic and non-parasitic samples" should be "Host plants selection of Centranthera grandiflora Benth. and nontargeted metabolomics analysis of its parasitic and non-parasitic samples"

   "the Regional Grant of National Natural Science Foundation of China (Grant number 32360098); Youth Grant of National Natural Science Foundation of China (Grant number 32001684); Key Project of Yunnan Local University Joint Fund (Grant number 202201AN070036) and General Program of Yunnan Fundamental Research Projects (Grant number 202201AT070020)." 

5. We note that your Data Availability Statement is currently as follows: All relevant data are within the manuscript and its Supporting Information files.

Additional Editor Comments:

Please consider the reviewers comments and adjust them accordingly in the manuscript.

Reviewers' comments:

Reviewer's Responses to Questions

**Comments to the Author**

1. Is the manuscript technically sound, and do the data support the conclusions?

Reviewer #1: Yes

Reviewer #2: Yes

2. Has the statistical analysis been performed appropriately and rigorously? 

Reviewer #1: Yes

Reviewer #2: Yes

3. Have the authors made all data underlying the findings in their manuscript fully available?

Reviewer #1: Yes

Reviewer #2: Yes

4. Is the manuscript presented in an intelligible fashion and written in standard English?

Reviewer #1: No

Reviewer #2: No

5. Review Comments to the Author

Reviewer #1: I find this study interesting. However, I cannot fully assess the overall content of the manuscript due to the complexity of the results and discussion sections. The authors combine the results and discussion, followed by a separate discussion, which makes it difficult to follow. I suggest improving the presentation by clearly separating the results and discussion sections. The results section should exclusively focus on presenting the findings. Any specific methods or approaches used should be detailed in the methods section. Discussions on the significance of the results should be confined to the discussion section, connecting the overall findings.

Currently, I do not find the manuscript to be well-organized or clearly presented. I recommend that the authors enhance the overall quality of the writing. If you are considering a plant parasitic study, analyzing the chloroplast genome would be highly beneficial. I suggest comparing the chloroplast genomes of parasitic and non-parasitic plants within the same family. The chloroplast genome for the species in question is already available. It is known that due to the parasitic nature, the chloroplast genome is under reduced selective pressure, leading to degradation.

Hopefully, you will add this in all parts of manuscript.

Reviewer #2: This manuscript assesses the host compatibility for C. grandiflora and contrasts the metabolic profiles of parasitized versus non-parasitized C. grandiflora plants. Collectively, the findings contribute to a deeper understanding of C. grandiflora's growth dynamics, which is instrumental for guiding its cultivation and application strategies moving forward. Nonetheless, the manuscript's current presentation requires refinement in both writing quality and organizational structure to meet the publication criteria.

1. Line Numbers: Please include line numbers in the manuscript to enhance the navigability and efficiency of the review process.

2. Paragraph Clarification: On page 3, the secondary paragraph contains a passage that requires rephrasing for clarity and correction of typographical errors: "...2-year-old seedlings with soil ball perennial roots [4]. the leaves of ...".

3. Table 1 Correction: In Table 1, there is a typographical error in the column name; "Name of the possible host palnts" should be corrected to "Name of the possible host plants".

4. Figure and Table Legends: The legends for figures and tables should be expanded with more detailed descriptions to improve clarity and understanding.

5. Consistency in Formatting: Ensure consistency in the use of spaces around the degree symbol for temperature indications, such as "-20℃" or "25 ℃", throughout the manuscript.

6. Figure 2 Enhancement: The inclusion of a scale bar in Figure 2 may aid reviewers in better grasping the spatial context presented in the image.

7. Clarification Needed: On page 12, in the subhead "Differential analysis of parasitic (JS) and non-parasitic (WJ) C. grandiflora", specify the aspects in which the differences are being analyzed. And also check through the manuscript.

8. Figure Title Formatting: The use of parentheses in figure titles, as seen in Figures 4 and 5, is not recommended.

9. Capitalization Consistency: Inconsistencies in capitalization have been noted in various sections, such as "Organic oxygen" and "Lipids" on page 14, which should be corrected for a professional presentation.

10. Table Presentation: The design of most tables should be refined to achieve a higher standard of formality and visual appeal.

6. PLOS authors have the option to publish the peer review history of their article (what does this mean?). If published, this will include your full peer review and any attached files.

Reviewer #1: No

Reviewer #2: No

---

## [Author Response · Author response to Decision Letter 0]

10 Jul 2024

Dear editors and reviewers,

On behalf of my co-authors, we thank you very much for giving us an opportunity to revise our manuscript entitled “Host plants selection of Centranthera grandiflora Benth. and nontargeted metabolomics analysis of its parasitic and non-parasitic samples”. We appreciate editors and reviewers for your positive and constructive comments and suggestions on our manuscript. We have studied the comments carefully and have made corrections which we hope meet with approval. Revised portion are marked in red using the track changes mode in MS Word.

---

## [Decision Letter · Decision Letter 1]

6 Aug 2024

PONE-D-24-19588R1Title. Host plants selection of Centranthera grandiflora Benth. and nontargeted metabolomics analysis of its parasitic and non-parasitic samplesPLOS ONE

Dear Dr. Liu,

Thank you for submitting your manuscript to PLOS ONE. After careful consideration, we feel that it has merit but does not fully meet PLOS ONE’s publication criteria as it currently stands. Therefore, we invite you to submit a revised version of the manuscript that addresses the points raised during the review process.

We look forward to receiving your revised manuscript.

Kind regards,

Rajib Chowdhury, M.Sc.; MPH

Academic Editor

PLOS ONE

Additional Editor Comments:

Dear Authors, Kindly review the comments and suggestions made by the reviewers' and incorporate them in the manuscript and resubmit it.

Reviewers' comments:

Reviewer's Responses to Questions

**Comments to the Author**

1. If the authors have adequately addressed your comments raised in a previous round of review and you feel that this manuscript is now acceptable for publication, you may indicate that here to bypass the “Comments to the Author” section, enter your conflict of interest statement in the “Confidential to Editor” section, and submit your "Accept" recommendation.

Reviewer #1: (No Response)

Reviewer #2: All comments have been addressed

Reviewer #3: All comments have been addressed

2. Is the manuscript technically sound, and do the data support the conclusions?

Reviewer #1: (No Response)

Reviewer #2: Partly

Reviewer #3: Yes

3. Has the statistical analysis been performed appropriately and rigorously? 

Reviewer #1: (No Response)

Reviewer #2: Yes

Reviewer #3: I Don't Know

4. Have the authors made all data underlying the findings in their manuscript fully available?

Reviewer #1: (No Response)

Reviewer #2: Yes

Reviewer #3: Yes

5. Is the manuscript presented in an intelligible fashion and written in standard English?

Reviewer #1: (No Response)

Reviewer #2: Yes

Reviewer #3: Yes

6. Review Comments to the Author

Reviewer #1: Dear author,

I am not satisfied with the response to my comments. However you have addressed some other comments. So I will left the decision to editor.

Reviewer #2: (No Response)

Reviewer #3: The revised manuscript is well written, and the authors have adequately addressed the previous comments raised.

7. PLOS authors have the option to publish the peer review history of their article (what does this mean?). If published, this will include your full peer review and any attached files.

Reviewer #1: No

Reviewer #2: No

Reviewer #3: **Yes: **Gerald Mboowa

---

## [Author Response · Author response to Decision Letter 1]

2 Sep 2024

This article represents a comprehensive study on the cultivation and metabolic aspects of the rare ethnic medicinal plant, Centranthera grandiflora Benth., in Yunnan Province. To the best of our knowledge as the authors, the content herein is deemed exhaustive and well-rounded. This plant is on the verge of extinction in China and the world, due to the fact that no artificial cultivation method has been found for a long time, and it is also classified as a rare plant in China. Our research team has spent nearly a decade uncovering the hemiparasitic nature of the Centranthera grandiflora Benth., thereby enabling its successful artificial cultivation. Currently, there is limited research on the growth and metabolism of hemiparasitic plants, making this article a highly valuable piece for publication and sharing. The article could provide some references for exploring the growth and metabolism mechanisms of hemiparasitic plants. However, although our team has achieved the artificial cultivation of Centranthera grandiflora Benth., the cultivation methods still require further refinement to facilitate large-scale planting and subsequently enable more scientific researches, which is a long-term work.

We have diligently addressed each and every comment from the reviewers, and the subsequent responses from the new set of reviewers have not surfaced any specific concerns or issues with the manuscript. Therefore, we humbly request the editor's final scrutiny to review our revised submission and kindly inform us of any necessary revisions or adjustments that may be required. Your invaluable insights and guidance would be greatly appreciated. Thank you for your attention and efforts in this regard.

---

## [Editor Report · Decision Letter 2]

5 Sep 2024

Title. Host plants selection of Centranthera grandiflora Benth. and nontargeted metabolomics analysis of its parasitic and non-parasitic samples

PONE-D-24-19588R2

Dear Dr. Liu,

We’re pleased to inform you that your manuscript has been judged scientifically suitable for publication and will be formally accepted for publication once it meets all outstanding technical requirements.

Kind regards,

Rajib Chowdhury, M.Sc.; MPH

Academic Editor

PLOS ONE
---

## [Editor Report · Acceptance letter]

18 Sep 2024

PONE-D-24-19588R2 

PLOS ONE

Dear Dr. Liu, 

I'm pleased to inform you that your manuscript has been deemed suitable for publication in PLOS ONE. Congratulations! Your manuscript is now being handed over to our production team.

Kind regards, 

on behalf of

Dr. Rajib Chowdhury 

Academic Editor

PLOS ONE